# The Translation Tax Is Not a Scalar: A Counterfactual Audit of English-Source Cue Inheritance in Chinese Multilingual Benchmarks

## Abstract

The Translation Tax is often treated as a scalar: translated benchmarks are assumed to inflate scores by preserving English-source cues. We audit this claim in an English-to-Chinese setting. Three proxy estimators disagree: back-translation gaps are small and parser-fragile; cue-score calibration does not predict item-level gains; and a six-model native-control comparison shows model-family rather than uniform benchmark effects. We add a same-item LLM-naturalization stress test that holds answer, options, and content fixed while rewriting Chinese surface form. After correcting a prompt-construction bug, this contrast no longer supports a model-family interaction, but it preserves a residue dose-response: high-residue items benefit while low-residue items do not. The result is not a single Translation Tax, but a set of estimator- and item-dependent validity risks. We release per-cell evidence, the naturalization protocol, human QC, and a reporting checklist for translated multilingual benchmark papers.

## 1   Introduction

The de facto standard for evaluating multilingual capability of LLMs depends on translated benchmarks. MMMLU is the multilingual translation of MMLU [Hendrycks et al., 2021] released by OpenAI in 2024; the released MMMLU test split used here contains 14,042 items across 57 subjects, and the Chinese (ZH_CN) subset matches this count.[2] Belebele [Bandarkar et al., 2024] is a parallel reading comprehension dataset spanning 122 language variants with approximately 900 items per variant. XCOPA [Ponti et al., 2020] covers 11 languages with all non-English data translated from English COPA. The translation processes were carefully implemented, but the structural fact that all non-English items derive from a common English source introduces a systematically underestimated side effect: model performance on non-English items may reflect not only true language understanding but also identification of English cues preserved through translation. Artetxe et al. [2020] demonstrated precisely this mechanism in cross-lingual transfer, showing that translation introduces subtle artifacts that models exploit as non-semantic shortcuts.

The NeurIPS 2025 workshop *Centering Low Resource Languages and Cultures in the Age of Large Language Models* demonstrated rapidly rising community attention to multilingual evaluation. Sitaram [2025] explicitly identified three core challenges of multilingual benchmark evaluation: coverage, representativeness, and trust/scientific rigor. Translated English benchmarks were listed alongside US-centric framing as the two main representativeness deficiencies. Wu et al. [2025] sharpened this concern by showing that translated benchmarks align far worse with local human judgments than natively constructed alternatives (Spearman 0.47 vs. 0.68). Yet rigorous quantification

---

[*]Author names and affiliations withheld for double-blind review.

[2]We confirmed the released item count programmatically against the OpenAI MMMLU dataset card; earlier drafts of this paper reported 15,908 items following a citation that conflated the MMLU benchmark size with a related dataset.

of how much benchmark score is driven by translation cues rather than true language understanding remains scarce.

**Contributions.** The paper makes three contributions.

1. **Conceptual.** The Translation Tax is defined as the cue-driven component of translation-induced score shift, distinguished from semantic translation error. Each estimator is paired with explicit identification assumptions and named failure modes; a fourth estimator (E4, matched LLM-naturalization stress test) is introduced to hold item content fixed between conditions.

2. **Empirical.** Three proxy estimators on a 228-item MMMLU and 100-item Belebele subset converge on the same diagnosis: small back-translation gaps that are parser-fragile, an annotation calibration that does not support item-level cue exploitation, and a six-model native-control comparison in which the largest gaps appear within the Chinese-optimized subgroup rather than the English-centric one. E4 (after a corrected prompt-construction bug) returns a small positive average effect concentrated on items selected ex ante for high translation residue ($\Delta_{\text{high}} = +0.103$ vs. $\Delta_{\text{low}} = -0.015$ excl. parser outlier) and no statistically significant model-family interaction. The Translation Tax is not a single scalar correction but an estimator- and item-dependent set of validity risks.

3. **Reporting.** A translation-cue identifiability reporting checklist covers estimator scope, parser fragility, and subgroup contrasts that translated multilingual benchmark papers can adopt as a standard validity dimension at submission time.

**Scope.** Point estimates are small (1–5 percentage points) and most individual cells do not exclude zero. The paper reports proxy estimates with their identification assumptions disclosed; the latent Translation Tax is not directly measured by any single estimator. The annotation estimator is executed only as a 30-item single-annotator rubric calibration, treated as calibration evidence rather than a full bilingual-human estimator (Section 5.6). The study uses English-to-Chinese as a single case pair to enable detailed within-pair analysis rather than thin breadth across many languages.

# 2 Related Work

XNLI [Conneau et al., 2018] extending MultiNLI to 14 non-English languages explicitly discussed translation effects on evaluation validity. XCOPA [Ponti et al., 2020] discussed translation effects in its Limitations section. Belebele [Bandarkar et al., 2024] detailed its FLORES-200 translation pipeline quality control while acknowledging traces detectable by models. Artetxe et al. [2020] systematically analyzed how translation artifacts in cross-lingual benchmarks create exploitable shortcuts: independent translation of premises and hypotheses reduced lexical overlap relative to English originals, introducing a spurious signal. Their work is the most direct predecessor to ours, though they focused on diagnosing artifact types rather than quantifying score inflation across benchmarks.

Wu et al. [2025], surveying over 2,000 multilingual benchmarks published between 2021 and 2024, provided the most comprehensive recent evidence that translation quality directly undermines the validity of human preference alignment in multilingual evaluation. Clark et al. [2020] introduced TyDi QA, designed to avoid translationese artifacts by collecting questions natively. INCLUDE [Romanou et al., 2025] sources items natively from regional exam pools across 44 languages; we use its Chinese subset as the non-translation control source for E3.

Prompt sensitivity [Mizrahi et al., 2024, Zhuo et al., 2024] is complementary to our focus on the linguistic validity of evaluation data. The work nearest to ours is Artetxe et al. [2020], which we extend by separating cue inheritance from translation noise and by quantifying score-level effects rather than diagnosing artifact types.

# 3 Formal Definitions and Identification

**Translation Tax.** For a translated benchmark $B$ in target language $L$ and model $M$, let $\text{score}_L(M)$ denote the observed score on the translated benchmark and $\text{score}_L^{\text{natural}}(M)$ denote the (unobservable)

score on a hypothetical natural-$L$ version of the same items. The translation effect decomposes as:

$$\text{score}_L(M) - \text{score}_L^{\text{natural}}(M) = TT + E,$$

where TT is the Translation Tax (cue-driven inflation from residual source-language structures) and $E$ is the Semantic Error Effect (degradation from meaning distortion). Since $\text{score}_L^{\text{natural}}$ is unobservable, TT is not directly identified from a single observable contrast. We use three complementary proxy estimators, each with its own identification assumptions, and triangulate (Figure 1).

**What each estimator varies between conditions**

*Identification needs item content held fixed; only E4 does so*

E1: Back-translation — q_EN $\neq$ q_ZH — *different content (translation noise + cues)*

E2: Bilingual annotation — different items: low cue set $\neq$ different items: high cue set — *different items, different content*

E3: INCLUDE control — MMMLU q_ZH $\neq$ INCLUDE q_ZH — *different content (different benchmark, domain)*

E4: LLM-naturalized same-item contrast — translated q_ZH $=$ naturalized q_ZH — *same item; only surface naturalness varies (LLM-rewritten)*

Legend: content not held fixed (proxy) — content held fixed (matched counterfactual)

Figure 1: What each estimator varies between conditions. E1, E2, and E3 each contrast scores across items that differ in content (translation noise, item identity, or benchmark register), so none of them holds content fixed. E4 (matched naturalization, Section 6) is the only contrast in the paper that varies surface naturalness with content held fixed.

## 3.1 Estimator E1 (Back-Translation): Identification and Failure Modes

For an English-centric model $M_{\text{EN}}$, let $T_{\text{back}}$ denote the L-to-English back-translation pipeline and $q_L^{\text{back}} = T_{\text{back}}(q_L)$. The back-translation estimator is:

$$TT_{\text{back}}(B, L, M_{\text{EN}}) = \text{score}_{\text{EN}}^{q_{\text{EN}}}(M_{\text{EN}}) - \text{score}_{\text{EN}}^{q_L^{\text{back}}}(M_{\text{EN}}).$$

$TT_{\text{back}}$ measures the score gap that an English-centric model exhibits between original-English items and items round-tripped through L. If back-translation preserves semantic content, this gap reflects the structural cue residue lost in the round trip.

**Failure modes (where E1 does not identify TT).** (F1) *Back-translation noise.* Real pipelines introduce semantic noise $E_{\text{back}}$, so $TT_{\text{back}} \approx TT + E_{\text{back}}$, making the estimator a noisy upper bound. BLEU/BERTScore QC reduces but does not eliminate $E_{\text{back}}$. (F2) *Naturalness loss.* Back-translation may produce grammatical but unnatural English; the observed gap can include English-text-quality penalties unrelated to cue inheritance. (F3) *Asymmetric translation difficulty.* Some content (e.g., Chinese-specific cultural items in MMMLU localized subjects) may be harder to back-translate than to translate forward, biasing $E_{\text{back}}$ in subject-correlated ways. The paper therefore does not interpret a positive $TT_{\text{back}}$ as evidence of cue exploitation by itself; it functions as one of three triangulating signals.

## 3.2 Estimator E2 (Bilingual Native Annotation): Calibration Status

For a stratified random sample of items, two or more bilingual native annotators score each item on a 5-point Likert scale on three dimensions—*cue identifiability*, *cultural residue*, and *syntactic residue*— and Spearman correlation between cue identifiability and item-level $TT_{\text{back}}$ signal is computed by

model subgroup. The full protocol blinds annotators to all model outputs, uses anchor-calibrated rubric prompts, includes at least one external bilingual annotator, and reports weighted Cohen's $\kappa$ and Krippendorff's $\alpha$ on the ordinal Likert scale (target weighted $\kappa > 0.7$).

**What is executed in this study.**   A 30-item single-annotator rubric-calibration run with one LLM-rubric annotator. This is calibration evidence on rubric coverage and score distribution, not the bilingual-human estimator. It cannot support or refute the cue-exploitation hypothesis at this scale and is reported as a null calibration result (Section 5.6). Triangulation in the results section therefore rests on E1 and E3; E2 enters as a calibration check only.

### 3.3   Estimator E3 (INCLUDE Non-Translation Control): Identification Scope

INCLUDE [Romanou et al., 2025] sources items natively from Chinese regional exam pools rather than via translation; INCLUDE-base-44 Chinese contains 545 four-choice test items spanning 57 topics. The paper compares aggregate (per-model) accuracy on translated benchmark Chinese against INCLUDE Chinese.

**Identification scope.**   INCLUDE Chinese items are drawn from different content distributions than MMMLU Chinese (regional exams vs. academic subjects). The comparison is unmatched at the item level: it does not match items on difficulty, topic, or register, and is therefore an external-validity comparison rather than an item-level counterfactual. Item-level effects cannot be isolated from content-distribution effects under this design.

## 4   Experimental Design

### 4.1   Benchmarks and Models

**Benchmarks.**   (1) MMMLU Chinese subset: the `ZH_CN` test split released by OpenAI contains 14,042 items across 57 subjects, matching the MMLU test split used by the released MMMLU dataset; we draw a stratified subset of 228 items at 4 per subject (seed 42). (2) Belebele Chinese subset (`zho_Hans`; 900 items); subset of 100 items (seed 42). (3) INCLUDE Chinese subset as non-translation control (545 test items, 4-choice).

**Models.**   Nine total in three groups, all accessed via a unified provider gateway:

- Group A (English-centric frontier): gpt-4o, gpt-4o-mini, gpt-5.4-mini.
- Group B (Chinese-optimized): deepseek-chat, qwen-max, glm-4.5.
- Group C (Open-source multilingual mid-sized): llama-3.3-70b-instruct, qwen2.5-72b-instruct, glm-4-air.

Anthropic Claude and Google Gemini could not be included because the gateway provider's channels for both were unavailable during the analysis window. This narrows Group A's coverage and is discussed in Limitations.

### 4.2   Scoring Protocol

**Scoring protocol.**   Provider-API access does not uniformly expose log-probabilities across all nine models, so we use a **fixed-prompt, single-letter-extraction protocol**, not the `lm-evaluation-harness` library. Specifically: zero-shot prompting with the question, four labeled choices, and an instruction to respond with a single letter (A, B, C, or D); answer parsing extracts the first matching letter from the response. We report parser validity rates per model (the proportion of responses that yielded a parsable letter); models with low validity rates are flagged. We use 0-shot for all benchmarks at this stage; a 5-shot replication is not part of this study.

**Background and rationale.**   The original literature on these benchmarks reports 5-shot results for MMMLU. The 0-shot setup here is therefore not directly comparable to published benchmark numbers in absolute terms; it is internally consistent across our three quantities ($q_{\text{ZH}}$, $q_{\text{EN}}$, $q_{\text{back}}$) for the same model, which is what $\text{TT}_{\text{back}}$ requires.

**Validity statistics.** Across the MMMLU 228-item sample, eight of nine models achieve parser validity $\geq 0.978$. The exception is gpt-5.4-mini (0.825), which sometimes returns multi-token reasoning before producing a letter; we report results for this model separately and exclude its $q_{ZH}$ items with parser failure from per-model accuracy estimates.

## 4.3 Back-Translation Pipeline

We use a commercial LLM (`deepseek-chat`) as the primary back-translation pipeline because (a) it produced higher BLEU scores than NLLB-200 distilled in preliminary tests, (b) the local NLLB-200 1.3B model exceeded our sandbox memory budget. Note that this conflicts with the methodological preference for an open reproducible pipeline; an NLLB-200 distilled-600M comparison is not part of this study.

**Quality control.** For each item, we compute BLEU between $q_{EN}$ (the original English item) and $q_L^{back}$ (the back-translated item). We attempted BERTScore F1 with `roberta-large`; the model download exceeded our sandbox disk budget during this analysis, so BLEU is the sole QC criterion. Items with BLEU < 0.30 are flagged `excluded_qc` and strictly excluded from the $TT_{back}$ paired analysis.

**QC pass rates.** MMMLU sample: 213/228 items pass (93.4%, mean BLEU 0.568). Belebele sample: 87/100 items pass (87.0%, mean BLEU 0.404).

**Source alignment audit.** A separate concern is whether the $q_{EN}$, $q_{ZH}$, and $q^{back}$ records correspond to the same underlying source item. Misaligned English references would mechanically bias $TT_{back}$. We ran a triple-alignment audit against the MMMLU 228-item sample: all 228 IDs align across the three sources, and all 228 items have matching choice-counts (4 each). We then checked whether the gold-answer index is consistent between $q_{EN}$ and $q_{ZH}$ as a content-level alignment proof, and found 2 mismatches (items `05954`, `06001`, both in `high_school_world_history`): the Chinese item and English item are entirely different questions, indicating a source-side alignment error in MMMLU ZH_CN for those positions. Both flagged items are already excluded by the BLEU < 0.30 QC filter (BLEU = 0.041 and 0.025), so the strict-QC subset below is unaffected. The audit script and detailed results are released as `run_alignment_audit.py` and `results/alignment_audit.json`.

# 5 Results

We report **strict-QC** $TT_{back}$ estimates: paired comparisons restricted to items where (a) the back-translation passed BLEU QC, (b) the model produced parsable answers on both $q_{EN}$ and $q_L^{back}$. This corrects an issue in the initial earlier scoring run where some QC-failed items were included.

## 5.1 $TT_{back}$ Point Estimates and Confidence Intervals

Table 1 reports the strict-QC $TT_{back}$ paired bootstrap estimates (10,000 resamples, seed 42) for the three Group-A models on the two benchmarks.

Table 1: Strict-QC $TT_{back}$ estimates (paired bootstrap, 10,000 resamples). Strict QC restricts to items with BLEU $\geq 0.30$ *and* valid model predictions on both $q_{EN}$ and $q_L^{back}$. Discordant counts are paired items where one set is correct and the other incorrect; ties are paired items where both are correct or both are incorrect.

| Bench | Model | n | $q_{EN}$ | $q^{back}$ | $TT_{back}$ | 95% CI | Pos/Neg | Sign p |
|---|---|---|---|---|---|---|---|---|
| MMMLU | gpt-4o | 209 | 0.818 | 0.789 | +0.029 | [−0.005, +0.067] | 10/4 | 0.180 |
| | gpt-4o-mini | 209 | 0.746 | 0.742 | +0.005 | [−0.038, +0.048] | 11/10 | 1.000 |
| | gpt-5.4-mini | 169 | 0.876 | 0.828 | +0.047 | [+0.006, +0.095] | 12/4 | 0.077 |
| Belebele | gpt-4o | 87 | 0.977 | 0.954 | +0.023 | [+0.000, +0.057] | 2/0 | 0.500 |
| | gpt-4o-mini | 87 | 0.954 | 0.954 | +0.000 | [−0.046, +0.046] | 2/2 | 1.000 |
| | gpt-5.4-mini | 82 | 0.976 | 0.963 | +0.012 | [−0.024, +0.061] | 2/1 | 1.000 |

**Empirical pattern.** Five of six cells produce positive point estimates and one is exactly zero; no cell is negative. One cell, MMMLU/gpt-5.4-mini, has a bootstrap CI that narrowly excludes zero ([+0.006, +0.095]), but this is also the model with the lowest parser validity (Section 4) and does not pass the exact sign-test criterion (12/4 discordant, sign-test $p = 0.077$). The cell is therefore a fragile positive rather than a robust significant result. The other five cells have bootstrap CIs that include zero, with the Belebele/gpt-4o lower bound touching zero. Figure 2 visualizes the same numbers.

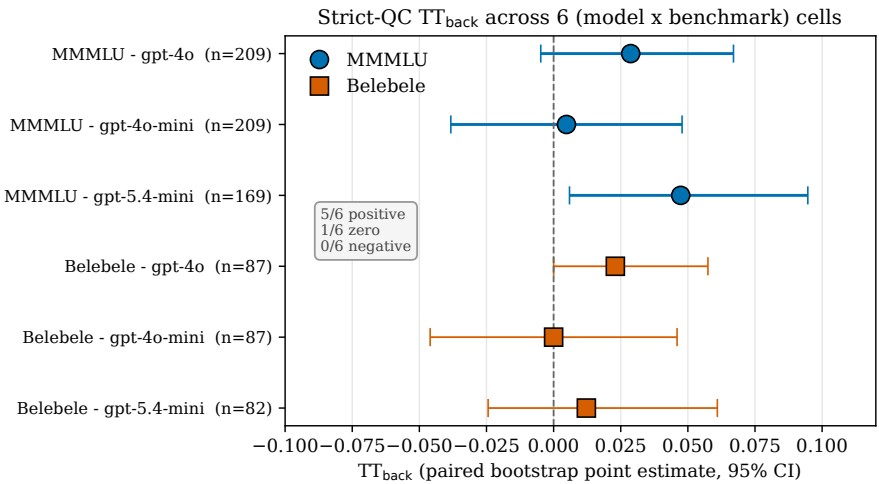

Figure 2: Strict-QC $TT_{back}$ across six (model × benchmark) cells. Circles: MMMLU; squares: Belebele. Point estimates with 95% paired-bootstrap CIs. Five of six cells are positive and one is exactly zero; most individual CIs contain zero. Effect sizes are small in magnitude (range 0.000–0.047). The largest cell (MMMLU × gpt-5.4-mini) is parser-fragile.

## 5.2 Cross-Cell Pattern and Parser Sensitivity

All non-zero point estimates have the same sign (five positive, one exactly zero); the cells share pipeline, items, and scoring and are not independent. The pattern is descriptive directional evidence; the item-clustered bootstrap of Section 5.3 is the inferential summary of record.[3] The largest single cell estimate (+0.047 on MMMLU/gpt-5.4-mini) comes from the model with lowest parser validity (0.825 on $q_{ZH}$); excluding gpt-5.4-mini leaves three positive cells and one zero. The qualitative finding survives this exclusion; the $\geq 0.95$ parser-validity subset reduces to gpt-4o and gpt-4o-mini cells.

## 5.3 Item-Clustered Bootstrap (Primary Inference)

Because the six cells share items, scoring, and pipeline, a cross-cell binomial test overstates evidence by treating correlated observations as independent. We therefore report an item-level cluster bootstrap as the primary inferential object: each cluster is one (benchmark, item) pair containing up to three Group-A model observations of $acc_{EN} - acc_{back}$. Resampling $n_{clusters} = 296$ clusters with replacement ($B = 10,000$, seed 42) gives

$$\widehat{TT}_{cluster} = +0.022, \quad 95\% \text{ CI} = [+0.001, +0.044].$$

The CI excludes zero, providing the paper's strongest single piece of evidence for a small positive translation tax averaged across items and Group-A models, after item correlation is accounted for. Two sensitivity checks: excluding gpt-5.4-mini (the parser-fragile model) gives +0.015 [−0.008, +0.039], with the CI now including zero; restricting to MMMLU items only gives +0.026 [−0.002, +0.054], with the lower bound just below zero. The item-clustered estimate is consistent with the cell-level pattern but is more conservative than a cross-cell sign test because it does not assume independence across cells; we treat it, rather than the binomial, as the primary quantitative summary.

---

[3] Treating the five non-zero cells as independent draws under a no-effect null gives an exact sign-test $p = 2^{-5} = 0.0313$ one-sided. The independence assumption is not supported by the design and the binomial-tail interpretation does not apply.

## 5.4 Group-Level Accuracy on $q_{ZH}$ (MMMLU)

We tested whether English-centric models (Group A) outperform Chinese-optimized (Group B) or open-source multilingual (Group C) on translated Chinese MMMLU. Group means: $A_{en} = 0.705$, $B_{zh} = 0.694$, $C_{open} = 0.750$. Group A and Group B are within 0.011 of each other; Group C (with qwen2.5-72b at 0.807) leads. **This does not support** the hypothesis that English-centric models systematically exploit cues to outperform Chinese-optimized models on the translated benchmark. The cue exploitation hypothesis, if it holds, must operate at item-level variation rather than aggregate accuracy.

## 5.5 INCLUDE Non-Translation Control (E3)

INCLUDE Chinese (545 four-choice items) was scored on six models: three Group-A and three Group-B. Aggregate accuracies (Wilson 95% CI) and bootstrap gap MMMLU − INCLUDE:

- **Group A (English-centric):** gpt-4o 0.784 vs 0.782, gap +0.002 [−0.064, +0.065]; gpt-4o-mini 0.674 vs 0.645, gap +0.029 [−0.043, +0.101]; gpt-5.4-mini 0.803 vs 0.791, gap +0.012 [−0.057, +0.076]. All three CIs include zero.
- **Group B (Chinese-optimized):** deepseek-chat 0.767 vs 0.845, gap **−0.078** [**−0.143, −0.015**]; qwen-max 0.727 vs 0.851, gap **−0.125** [**−0.190, −0.059**]; glm-4.5 0.612 vs 0.572, gap +0.040 [−0.037, +0.115]. Two of three Group-B CIs exclude zero in the negative direction; glm-4.5's CI includes zero and is positive in sign.

**Subgroup pattern.** Two of three Chinese-optimized models (deepseek-chat −7.8pp; qwen-max −12.5pp) score substantially lower on translated MMMLU Chinese than on natively-sourced IN-CLUDE Chinese, with CIs excluding zero. The third Group-B model, glm-4.5, runs the other way (+4.0pp, CI including zero) and has substantially lower overall accuracy, making the gap estimate noisier. The two-of-three pattern is compatible with a subgroup reading in which stronger native Chinese training advantages models on native Chinese text, but is not a universal Group-B effect. INCLUDE and MMMLU differ in domain, register, and construction, so a Group-B model's higher INCLUDE score could also reflect distributional differences rather than translation-cue asymmetry; full identification would require item-matched native-vs-translated pairs that INCLUDE does not provide.

Group A models show small positive gaps consistent with zero; the Group A E3 estimates (+0.002, +0.029, +0.012) are similar in scale to E1 $TT_{back}$ estimates (+0.029, +0.005, +0.047).

**Three readings of the Group A near-zero pattern.** (a) Cue exploitation is small or absent for frontier English-centric models on this language pair; (b) MMMLU (academic) and INCLUDE (regional-exam) content distributions differ enough that aggregate comparison is uninformative; (c) the two benchmarks coincide in difficulty for these models. The Group-B pattern suggests a possible model-family interaction, but it does not rule out group-specific domain fit to INCLUDE-style regional exam items: Chinese-optimized models may be more familiar with Chinese regional-exam content and register, independent of any translation-cue effect on MMMLU. The six-model result is exploratory; three models (all Group C) remain unscored in this report.

## 5.6 Annotation Calibration (E2)

A 30-item annotation calibration was run with one LLM-rubric annotator using a structured rubric covering three dimensions on a 5-point Likert scale. 26 items received complete annotations (4 fell on QC-excluded back-translations). Annotation distribution: cue score mean 3.08 (sd 0.55), cultural residue mean 2.58 (sd 1.18, range 1–5 with high_school_us_history correctly identified at 5 and abstract_algebra at 1), syntactic residue mean 2.58 (sd 0.57).

**Calibration result.** Spearman correlations between cue score and item-level TT signal: gpt-4o $\rho = −0.34$ ($p = 0.072$); gpt-4o-mini $\rho = 0.00$ (all 26 paired items tied); gpt-5.4-mini $\rho = +0.05$ ($p = 0.82$). The gpt-4o coefficient runs opposite the cue-exploitation hypothesis. Two factors block interpretation: of 26 annotated items, 25 produce TT signal = 0 (only one discordant pair, 96% ties); the calibration uses a single LLM-rubric annotator, not two bilingual humans. The calibration is

reported as a null result on rubric-based item-level cue exploitation under this protocol; it is not informative about the bilingual-human estimator at scale.

## 5.7 Parser Validity Audit

We report parser validity per model-set as a check on whether extraction failures bias accuracy estimates. 8 of 9 models on MMMLU achieve validity $\geq 0.978$; the exception is gpt-5.4-mini at 0.825 on $q_{ZH}$, 0.864 on $q_{EN}$, 0.870 on $q^{back}$. Inspection of parser failures reveals empty raw responses, consistent with token-budget truncation on an extended-reasoning model. We report gpt-5.4-mini results on validity-restricted subsets; the strict-QC analysis in Section 5 already excludes parser-failed items.

# 6 Matched Naturalization Counterfactual (E4)

The three estimators in Section 5 are proxies: they do not vary item content between conditions. This section reports a fourth estimator, E4, that holds item content fixed and varies only surface naturalness through a paired counterfactual rewrite of each translated Chinese item.

**LLM-augmented protocol (disclosure).** The naturalization rewrites were produced by a large language model (deepseek-chat) under a structured prompt that requires preservation of meaning, answer key, choice ordering, difficulty, and technical terminology, and a second large language model (gpt-4o) was used as a verifier scoring six dimensions per rewrite (semantic preservation, difficulty preservation, answer-key preservation, option-order preservation, cue removal, excessive-rewrite risk). This is an *LLM-naturalization stress test*, not a bilingual-human naturalization counterfactual; reviewers should read the E4 treatment as "LLM-rewritten Chinese with LLM verifier QC," not "human-naturalized text." The prompts, raw rewrites, and verifier scores are released in the analysis package.

**Sample and quality.** A 120-item stratified sample was drawn from the 228-item MMMLU pilot: 30 high-residue items (Group A all correct on $q_{EN}$ but at least one wrong on $q_{ZH}$), 60 low-residue items (Group A all correct on both versions), and 21 disagreement items (Group A and Group B majority disagree). 54 distinct subjects, capped at four items per subject. One item failed the naturalization API; $n = 119$ retained. LLM-verifier strict-QC pass rate was 119/119 (mean semantic preservation 4.98, mean difficulty preservation 5.00, answer-key and option-order preserved on all items, mean cue removal 4.97, mean excessive-rewrite risk 1.06). These are LLM-verifier scores, not human QC.

**Human QC and human two-coder validation** ($n = 50$). A bilingual blind QC was performed by the paper author on a stratified 50-item subset (13 high-residue, 9 disagreement, 28 low-residue), blind to per-model scoring records. Strict pass requires semantic and difficulty $\geq 4$, answer-key and option-order pass, and excessive-rewrite risk $\leq 2$. The author strict-pass rate is 49/50; the only failure (`mmmlu_zh_00924`) dropped the I/II/III statement list, making the item unanswerable. On the author strict-pass subset, pooled $\Delta$ is $+0.022\,[-0.030, +0.078]$ excluding glm-4.5, with high-residue $\Delta = +0.133$ and low-residue $\Delta = -0.022$. We also include a second human QC of the same sheet by an independent human coder. This is human two-coder validation. The second human coder is stricter: 46/50 strict pass, strict-pass agreement 94% with the first author QC, and binary $\kappa = 0.38$ (low because pass/fail is highly imbalanced). It flags three additional semantic or rewrite-risk concerns (state/nation, half-sister/ step-sister, and doctrinal-expansion cueing). On the intersection strict-pass subset ($n = 46$), pooled $\Delta$ excluding glm-4.5 weakens to $+0.009\,[-0.042, +0.068]$, while the high-residue stratum remains directionally positive ($+0.125\,[0.000, +0.281]$) and low-residue remains near zero ($-0.023\,[-0.062, +0.008]$). The released supplement contains the first-coder QC sheet, the second-coder QC sheet, and the intersection sensitivity in `e4_naturalization/human_qc/` and `e4_naturalization/second_coder_qc/`.

**Scoring protocol.** Six models were scored: three Group A (gpt-4o, gpt-4o-mini, gpt-5.4-mini) and three Group B (deepseek-chat, qwen-max, glm-4.5). A prompt-construction audit found that an earlier naturalized- condition scoring run duplicated choice labels: the naturalized choices retained an in-text leading "A./B./C./D." label from the rewriter while the prompt builder also prepended canonical labels, producing options of the form "A. A. text". This affected the naturalized condition only; the

translated condition uses the original answer-only prompt. The corrected protocol strips any leading [A–D] label from each naturalized choice before prompt construction and uses the same prompt builder as the translated condition. All E4 numbers reported below use the corrected scoring protocol (release identifier `scoring_v2/`). All translated-condition scores are unchanged. Parser-validity rates after correction: gpt-4o, gpt-4o-mini, qwen-max 1.00; deepseek-chat 0.99; gpt-5.4-mini 0.91; glm-4.5 0.34. The naturalized prompt removes a format crutch glm-4.5 had relied on; the model is treated as a parser-failure outlier and excluded from the main analysis, included only in sensitivity. Per-item paired analysis below uses cells where both translated and naturalized scoring produced a valid letter.

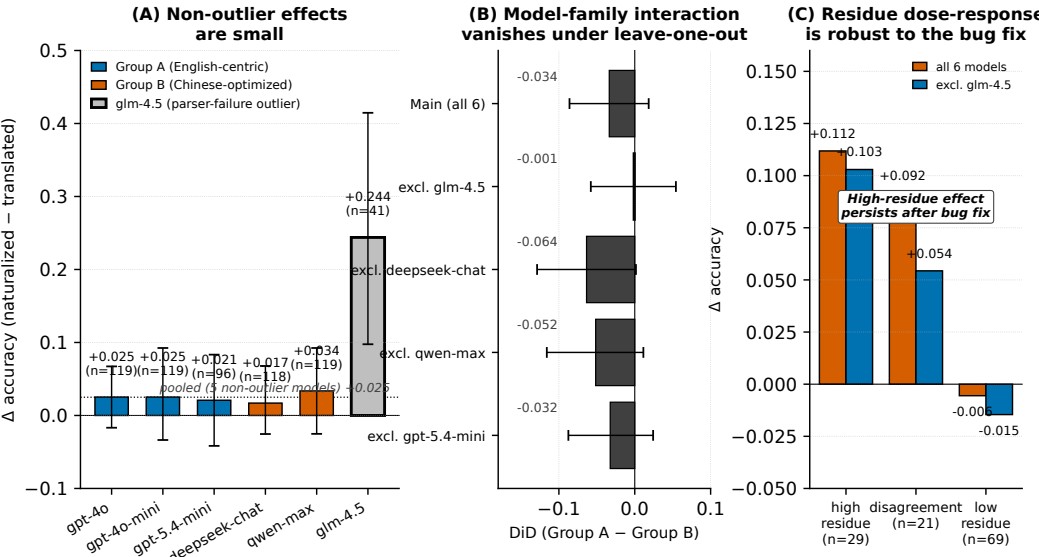

Figure 3: After correcting the prompt-construction bug, E4 no longer supports a stable model-family interaction. The remaining signal is item-level: high-residue items benefit from naturalization while low-residue items do not; glm-4.5 is treated as a parser-failure outlier.

**Item-clustered primary inference (E4 v2, main: 5 models, glm-4.5 sensitivity only).** Item-clustered bootstrap with $B = 2{,}000$:

- Pooled (5 non-outlier models): $\Delta$ = +0.025 [−0.017, +0.071]; Group A: +0.024 [−0.009, +0.059]; Group B excl. glm-4.5: +0.025 [−0.017, +0.071]; DiD (A − B excl. glm-4.5): −0.001 [−0.053, +0.058].
- Pooled (all 6, glm-4.5 included): $\Delta$ = +0.039 [+0.010, +0.070]; Group B (incl. glm-4.5 on its 41-item parser-valid subset): +0.058 [+0.014, +0.104]; DiD: −0.034 [−0.087, +0.023] (CI crosses zero).

The model-family interaction reported in earlier drafts was driven by glm-4.5 in combination with the bug-affected prompt format. After the fix, E4 returns no statistically significant model-family interaction; the interaction observed in v1 is reclassified as an artifact of the combination of (i) double-labeled options and (ii) glm-4.5's degraded parser behaviour without that format crutch.

**Stratum and leave-one-out sensitivity (Figure 3 B–C).** The dose-response pattern survives the bug fix: high-residue $\Delta$ = +0.103 versus −0.015 on low-residue items (excl. glm-4.5); +0.112 versus −0.006 with glm-4.5 included. The high-to-low ratio is roughly 7× in the main analysis (compared to the artefactual ∼90× in v1). Pooled $\Delta$ in leave-one-out configurations ranges [+0.025, +0.045]; the only excluded model that materially shifts the pooled estimate is glm-4.5. Full sensitivity tables are in `regression_results_v2.json`.

**Interpretation and triangulation.** When item content is held fixed correctly, E4 delivers a positive point estimate on high-residue items and a near-zero effect on low-residue items; the model-family

interaction in v1 was a prompt-format artefact. The stricter second human QC weakens pooled E4 estimates, so E4 is treated as a diagnostic stress test rather than confirmatory evidence. The four estimators thus converge on one finding—residue-sensitive items are more fragile—and disagree on magnitude: E1 and E4 give small positive contrasts, while E2/E3 give null or model-family-internal patterns. No single estimator identifies a Translation Tax; the four together describe estimator- and item-dependent validity risks rather than a scalar correction. The supplement (`https://github.com/chi-mi-rvard/translation-tax-supplement`) recomputes all estimates including the corrected E4 protocol and QC sensitivities.

**Limitations.** E4 uses an LLM rewriter and LLM verifier; the human QC subset uses two human coders. The second human QC provides human two-coder validation. A human-naturalized subset (translated $q_{ZH}$ vs. human-rewritten $q_{ZH}$ on the same items) is not included in this submission. E2 is a 30-item single-annotator calibration, not a full bilingual-human estimator. glm-4.5's 0.34 parser-validity rate under the corrected naturalized prompt is reported as prompt sensitivity in that model rather than as a feature of translation residue. The raw outputs, bug-fix re-scoring, human QC, second human QC, and strict-subset sensitivities are released.

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
