# OpenReview forum: "Auditing the Translation Tax: When Cross-Lingual Knowledge Transfer Distorts Benchmark Evidence"
_ijcai.org/IJCAI-ECAI/2026/Workshop/GENAIK-NORA — IJCAI-ECAI 2026 Joint Workshop on GENAIK and NORA_

### Official Review · Reviewer_xkDM · 2026-06-05
**Translatio**

**Rating:** 7
**Confidence:** 3

**Review:**

The paper studies Translation tax. Translation tax is the part of a translated benchmark’s score that comes from preserved source-language cues rather than true target-language understanding (as per my understanding).

Pros

* The paper addresses a real weakness in multilingual evaluation: translated benchmarks may not measure the same construct as native-language benchmarks. The motivation is well grounded in the introduction
* The paper explicitly separates cue-driven score inflation from semantic translation error
* The authors avoid overstating the six-cell sign pattern and correctly treat the item-clustered bootstrap as the primary inferential object because cells share items, models, and pipeline components. They also report sensitivity excluding the parser-fragile model.
* The paper does a good effort towards reproducibility

Cons
* The main experiments use only 228 MMMLU items, sampled as 4 per subject, and 100 Belebele items. The paper is clear about this, but the title and abstract still make a broad claim about “Chinese multilingual benchmarks” and the Translation Tax not being scalar
* The paper uses deepseek for backtransklation. However, deepseek-chat is also one of the evaluated chinese-optimized models, this creates potential model-family leakage and style coupling.
* The scoring protocol extracts the first matching letter from unconstrained model responses. This is fragile for reasoning-heavy or format-sensitive models. The paper itself reports low parser validity
* Although the paper is mostly clear and transparent, the term “Translation Tax” is overloaded. Sometimes it means cue-driven inflation; sometimes the evidence seems to measure general translation residue or translationese harm. The paper would be stronger if the terminology matched the signs of the empirical contrasts.
* The abstract says “three proxy estimators disagree,” but E2 is only a calibration check. This should be softened.

Other comments:
* There seems to be a slight mismatch between the title in the openreview and in the pdf. Please fix it.

---

### Official Review · Reviewer_LzQH · 2026-06-05
**This paper focuses on whether translated multilingual benchmarks contain residual English-language cues that can influence model performance. The problem is relevant and important and the proposed naturalization-based analysis is a useful contribution, but the paper is somewhat difficult to follow and could have been written well with simple terminology and several of the experiments are limited in scale or strength for example, the findings based on a single locale pair and smaller scale human annotation testing. Overall, I found the work interesting and suitable for workshop discussion, though the evidence is not strong enough to fully support conclusions broadly.**

**Rating:** 7
**Confidence:** 4

**Review:**

This paper studies whether translated multilingual benchmarks contain residual English-language cues that can influence model performance. The topic is timely and relevant, and the proposed naturalization-based analysis is an interesting contribution. I also appreciate the authors correcting a bug that affected an earlier version of the results.

The paper is harder to follow where it could have been written with a simple terminology and explaining the hypotheses well. The main contributions are introduced late, and is further complicated by multiple estimators and methodological details. The limitations of the paper are clearly mentioned. The study is limited to English - Chinese locale pair and a small annotation set of 30 examples and it will be interesting to see how the results/claims will vary for other locale pairs at a larger scale.

Overall, I found the work interesting and suitable for workshop discussion. While the analysis is not yet strong enough to fully support the overall claims, the paper does raise important questions about multilingual benchmark validity and provides useful directions for future work.

I got slightly confused by the inclusion of the NeurIPS submission notice and the full NeurIPS checklist in this workshop submission. This may likely be a formatting/template issue, but it would be helpful for the authors to clarify or remove conference-specific material that is not relevant here.

---

### Official Review · Reviewer_QUJU · 2026-06-06
**An Interesting Study of Translation Artifacts in Multilingual Evaluation, but with Limited Causal Evidence and Generality**

**Rating:** 4
**Confidence:** 4

**Review:**

+ Summary

This paper investigates the extent to which multilingual benchmark evaluation may be affected by translation artifacts. It introduces the concept of the Translation Tax, defined as cue-driven score inflation induced by residual source-language structures.
The authors propose a decomposition framework and estimate this effect using four complementary estimators (E1–E4): back-translation, bilingual annotation, native benchmark comparison, and an LLM-based naturalization counterfactual.
Empirically, the paper finds small, mostly positive effects, with stronger signals appearing in high-residue items rather than uniformly across benchmarks.
The main conclusion is that the Translation Tax is not a single scalar correction, but rather an estimator- and item-dependent set of validity risks.


+ Strengths

  + The paper formalizes the Translation Tax as a distinct component separated from semantic translation error. This conceptual decomposition is valuable and helps clarify a long-standing but previously loosely defined concern in multilingual evaluation.

  + The use of multiple proxy estimators (E1–E4) provides a broad empirical view of the problem. While each estimator has limitations, this triangulation improves robustness compared with relying on a single evaluation design.

  + The introduction of a matched-item counterfactual through LLM-based naturalization (E4) is interesting. The finding that effects concentrate in high-residue items is an important insight that goes beyond aggregate accuracy analysis.



+ Weaknesses

  + The most critical limitation is the absence of a symmetric evaluation across translation directions. The study focuses primarily on English-to-Chinese translation-derived benchmarks, without considering the reverse direction or comparable native-English versus translated-English settings. As a result, it is not possible to determine whether the observed effects are specific to English-source artifacts or reflect more general properties of translated text, such as translationese or unnatural distributions. This substantially limits the causal interpretability and generality of the conclusions.

  + While E3 (INCLUDE) attempts to introduce a non-translation control, the comparison is not matched at the item level and involves substantial differences in domain and distribution. Similarly, E4 relies on LLM-generated “naturalized” text rather than human-constructed native equivalents. Therefore, the study lacks a clean counterfactual in which content, difficulty, and distribution are controlled while only the translation factor varies.

  + Most reported effects seem to be small. Even the main item-clustered estimate becomes non-significant under reasonable sensitivity checks. This raises questions about the practical significance and robustness of the findings.

  + The four estimators do not fully agree:
    + E1 shows small positive effects.
    + E2 finds no correlation.
    + E3 shows model-dependent and inconsistent patterns.
    + E4 shows conditional effects concentrated in high-residue items.
  This limited cross-method consistency makes it difficult to conclude that a single underlying phenomenon is being captured.

  + The E4 estimator depends on LLM-based rewriting and verification, which may introduce potential biases such as imperfect semantic preservation, stylistic homogenization, or model-dependent rewriting artifacts. Although some human quality control is conducted, it is limited in scale and does not fully validate the counterfactual.

+ Comments

  + This paper lacks a dedicated Conclusion section, and the Limitations discussion is embedded at the end of Section 6 rather than presented as a standalone section. This seems to be problematic. (the NeurIPS checklist refers to “Section 8,” but no such section seems to exist in the manuscript.)

  + In addition, one of the stated contributions is a reporting checklist for translated multilingual benchmarks, but this checklist is not clearly presented in the main body. The reader is left unsure whether this is in the supplement or conflated with the NeurIPS checklist.

  + Section 6 is also overloaded: it combines method description, LLM naturalization protocol, human QC, bug correction, results, sensitivity analyses, interpretation, and limitations. The paper would benefit from separating E4 methodology, E4 results, discussion, limitations, and conclusion into distinct sections.

---

### Official Review · Reviewer_DBwQ · 2026-06-08
**A Good Paper Deconstructs the Translation Tax**

**Rating:** 7
**Confidence:** 4

**Review:**

This paper investigates the translation tax problem in the evaluation of multilingual large language models. The authors first introduce three complementary proxy estimators, then design the fourth estimator under the LLM-augmented protocol. Experimental results show that translation tax is an estimator- and item-dependent set of validity risks.

- Pros
1. The fourth estimator that LLM-naturalization stress test is interesting, which isolates the effect of translation.
2. The material related this paper will be benefit to the future work after public.

- Cons
1. Limited evaluation samples. The experiments are only conducted on English-Chinese pair, and subsets of MMMLU and Belebele. It may not generalize to other languages or multilingual benchmarks.
2. Over-reliance on LLMs. The estimator 4 relies on commercial LLMs for rewriting and verification, which risks bringing LLM-induced biases.

---

### Decision · Program_Chairs · 2026-06-10

Accept